# Densification of Two Forms of Nanostructured TATB under Uniaxial Die Pressures: A USAXS–SAXS Study

**DOI:** 10.3390/nano13050869

**Published:** 2023-02-26

**Authors:** Yan Zhou, Jing Shi, Mark Julian Henderson, Xiuhong Li, Feng Tian, Xiaohui Duan, Qiang Tian, László Almásy

**Affiliations:** 1State Key Laboratory of Environment-Friendly Energy Materials, School of Materials and Chemistry, Southwest University of Science and Technology, Mianyang 621010, China; 2Shanghai Synchrotron Radiation Facility, Shanghai Advanced Research Institute, Chinese Academy of Sciences, Shanghai 201204, China; 3Institute for Energy Security and Environmental Safety, Centre for Energy Research, 1121 Budapest, Hungary

**Keywords:** TATB, small-angle scattering, void, porosity, energetic material, explosive

## Abstract

Sequential ultra-small-angle and small-angle and X-ray scattering (USAXS and SAXS) measurements of hierarchical microstructure of a common energetic material, the high explosive 2,4,6-Triamino-1,3,5-trinitrobenzene (TATB), were performed to follow the microstructure evolution upon applied pressure. The pellets were prepared by two different routes—die pressed from a nanoparticle form and a nano-network form of TATB powder. The derived structural parameters, such as void size, porosity, and the interface area, reflected the response of TATB under compaction. Three populations of voids were observed in the probed *q* range from 0.007 to 7 nm^−1^. The inter-granular voids with size larger than 50 nm were sensitive to low pressures and had a smooth interface with the TATB matrix. The inter-granular voids with size of ~10 nm exhibited a less volume-filling ratio at high pressures (>15 kN) as indicated by a decrease of the volume fractal exponent. The response of these structural parameters to external pressures implied that the main densification mechanisms under die compaction were the flow, fracture, and plastic deformation of the TATB granules. Compared to the nanoparticle TATB, the applied pressure strongly influenced the nano-network TATB due to its more uniform structure. The findings and research methods of this work provide insights into the structural evolution of TATB during densification.

## 1. Introduction

In order to understand the processing–structure–property relationship of energetic materials, it is crucial to obtain a detailed description of their microstructures [1,2]. Scattering techniques, X-ray, and neutron scattering are versatile methods allowing researchers to resolve and quantify the hierarchical structural features on a length scale from ångström to micrometer. While wide-angle diffraction and total scattering methods provide information on the interatomic distances [3,4,5], the small-angle X-ray and neutron scattering (SAXS and SANS) techniques probe the microstructure of energetic materials, such as void size and morphology, inter- and intra-granular porosity, as well as the interfacial surface areas among the explosive particle, binder, and voids. Ultra-small-angle scattering combined with small-angle scattering can quantitatively reveal structural characteristics at larger length scales from 1 nm to 10 μm [5,6,7,8,9,10,11,12,13,14,15,16,17,18]. Compared with traditional techniques such as optical and electron microscopy, and X-ray imaging, the small-angle scattering is nondestructive, quantitative, and, when contrast variation is used, has the advantage of selectivity to the studied multiphase structures. Furthermore, it does not require complicated sample preparation, which is a benefit for characterizing explosive materials. The application of small-angle scattering in studies of energetic materials has been described in recent reviews [5,9]. 

TATB-based energetic materials are often used as the majority component of insensitive high explosive formulations, such as PBX-9502, PBX-9503. and LX-17 [8,10,11], due to their high detonation velocities and extreme safety. The thermal and shock stability of TATB is considerably greater than that of other known energetic materials. TATB crystal has a graphitic structure and displays weak van der Waals interaction along the *c*-axis [19], imparting a good plastic property and a good processability for TATB formulations. Prior to small-scale applications, TATB molding powder is usually die pressed into the form of a circular disc or cylinder. The microstructure of the compressed TATB solids is formed in a complex way as the function of the compression molding parameters [20,21]. Apart from the external parameters such as the applied pressure, temperature, and pressing method, the differences in the initial powder morphology, e.g., particle size, shape, and morphology, also affect the formed microstructure of the internal voids and the interfacial surface area. Ultra- and small-angle neutron scattering studies reveal the internal void fractal structure and preferred void orientation of uniaxially pressed PBX 9502 (95% TATB; 5% fluoropolymer binder) [22,23]. USAXS data obtained on TATB discs indicate the differences in surface properties (smooth, diffuse, or rough fractal-like) depending on the material lot and manufacturer [24]. X-ray computed tomography indicates that the multi-scale defects within TATB grains lead to inhomogeneous flow and deformation of TATB during die compression [25,26,27]. The complex and uneven microstructure of compressed TATB solids induce further, less predictable structural changes upon applied thermal and stress conditions [28,29,30]. So far, the structural evolution of TATB during compression and application is neither fully understood nor described, and quantitative experimental methods in microstructure determination are in demand.

The purpose of this work was to study the microstructure of TATB discs die pressed from two forms of nanostructured TATB powders. Sequential ultra-small-angle and small-angle X-ray scattering (USAXS and SAXS) measurements were combined to determine the size and number density of voids, porosity, and interface area as a function of applied pressures. The intra-granular voids were found to be sensitive to low pressure and had a smooth interface with the TATB host, while the inter-granular voids had a fractal network structure at high pressures. The response of the two forms of the TATB samples upon compaction were studied comparatively. The methods presented in this work can be used to support mesoscale simulations as well as to quantify lot-to-lot variations in the course of explosive production.

## 2. Materials and Methods

### 2.1. Materials

Two forms of nanostructured TATB powders were provided by the Institute of Chemical Materials (China Academy of Engineering Physics, Mianyang, China). The nanoparticle TATB, composed of isolated anisotropic particles with size of 50~500 nm (Figure 1a), was synthesized by solvent/nonsolvent recrystallization with concentrated sulfuric acid as the solvent and water as the nonsolvent [31]. The nano-network TATB, consisting of connected porous particles (Figure 1b), was prepared by high-energy ball milling of a TATB–water–surfactant mixture [32]. The nanostructured TATB powders were uniaxially die pressed to TATB discs (ø10 mm × 1.5 mm) with pressures of 1, 2, 5, 10, 15, and 30 kN. The TATB discs consolidated from nanoparticle form and nano-network form were denoted as NPD and NND, respectively. The bulk densities (*d*_bulk_) of the pressed TATB discs were calculated by their mass to volume ratios.

### 2.2. Characterization

USAXS measurements were recorded on an USAXS endstation located at BL10U1 beamline at Shanghai Synchrotron Radiation Facility. The wavelength (*λ*) of the incident X-rays was 0.124 nm. An Eiger 4M detector (Dectris, Baden, Switzerland) with a pixel size of 75 μm recorded the scattering intensity (*I*), which was measured as a function of scattering vector magnitude, *q* = 4πsin *θ*/*λ*, where *θ* is half of the scattering angle. The samples were placed at a distance of 27.6 m away from the detector, resulting in a *q* range of 0.007~0.15 nm^−1^.

Laboratory SAXS measurements were recorded on a SAXSpace instrument (Anton Paar, Graz, Austria) using Cu-*K_α_* irradiation. The wavelength *λ* of the X-ray was 0.154 nm. The scattering intensity was registered by a Mythen2 R 1K (Dectris, Switzerland) detector, which was placed at 317 mm from the sample. The samples were placed in designated sample holders and were kept in vacuum during the measurements. The measured *q* range was from 0.07 to 7 nm^−1^, and the recorded intensities were slit-smeared by the typical slit width of 1 cm. SAXS data were normalized to the incident primary beam intensity, corrected for background scattering, and were desmeared using the SAXSquant software (Anton Paar, Graz, Austria, version 4.1.0.7505). The measured intensities were converted to absolute cross-section unit by a calibration using pure water as standard. The USAXS data obtained on arbitrary scale were merged with the calibrated SAXS data, resulting in datasets with an overall *q* range from 0.007 to 7 nm^−1^ and absolute intensity with units of cm^−1^. Fitting of the USAXS–SAXS data to analytical model functions was performed by the method of least squares using SASfit software (Paul Scherrer Institute, Villigen, Switzerland, version 0.94.6) [33].

The cross-sections of the samples were observed by an EVO 18 scanning electron microscope (Carl Zeiss, Jena, Germany). Nitrogen sorption isotherms were measured by an Autosorb-iQ instrument (Quantachrome Instruments, Boynton Beach, FL, USA). The specific surface area was calculated by Brunauer–Emmett–Teller (BET) method. The samples were degassed at 100 °C for 7 h prior to the measurements. 

## 3. Results

### 3.1. Inter- and Intra-Granular Voids

The scattering data of the two series of TATB discs pressed at different pressures are shown in Figure 2. With the increase of applied pressures, for both NPD and NND samples, the deflection points of the plateau around 0.02 nm^−1^ shifted to high *q*, an indication of a decrease of the void size; and the slope decreased in the linear part of the profile at the high *q* values from 0.5 to 4 nm^−1^. Therefore, the intensity curves showed a hierarchical structure with two types of morphologies at typical sizes ≈50 nm and 10 nm, as determined by the *q* values of the apparent curvatures. A Kratky plot of the USAXS–SAXS data obtained from the samples die pressed at 30 kN is shown in Figure 3. Both the curves showed a broad peak in the *q* range of 0.01 nm^−1^ to 0.1 nm^−1^, corresponding to the scattering of the big voids. Compared to the NPD sample, the broad peak of the NND sample appeared at higher *q*. This difference indicates that the void size of the NND sample is smaller than that of the NPD sample. 

This kind of hierarchical structure can be conveniently modelled by Equations (1) and (2), as proposed by Hammouda [34], where the scattering intensity is described by the sum of Guinier equations and generalized Porod laws:(1)$$\left\{ \begin{array}{ll} I{\left(q\right)}_{11}=G_{1}\exp(-q^{2}R_{g1}^{2}/3)\; & \mathrm{for}\;q\leq q_{1}^{\prime }\\ I{\left(q\right)}_{12}=D_{1}/q^{d1}\; & \mathrm{for}\;q\geq q_{1}^{\prime } \end{array}\right.$$
(2)$$\left\{ \begin{array}{ll} I{\left(q\right)}_{21}=G_{2}\exp(-q^{2}R_{g2}^{2}/3)\; & \mathrm{for}\;q\leq q_{2}^{\prime }\\ I{\left(q\right)}_{22}=D_{2}/q^{d2}+B_{g}\; & \mathrm{for}\;q\geq q_{2}^{\prime } \end{array}\right.$$
where *I*(*q*)_11_ and *I*(*q*)_12_ appear at low *q*; *I*(*q*)_21_ and *I*(*q*)_22_ at high *q*. The parameters *R*_*g*1_ and *R*_*g*2_ are the radii of gyration of the featured size of big and small voids; *G*_1_ and *G*_2_ are prefactors of the Guinier law, and *D*_1_ and *D*_2_ are scaling factors of the power law term. The constant *B_g_* accounts for the background signal; and *d*_1_ and *d*_2_ are the exponents of the Porod terms. The continuity of the scattering intensity curves in the Guinier and Porod terms in Equations (1) and (2) defines the values of *q*_i_′ and *D*_i_′ [34].

To accomplish a reliable fitting in terms of the nine variables in Equations (1) and (2), the following strategies were combined: (i) fit the low *q* region of 0.02 nm^−1^ to 0.2 nm^−1^, which dominates the scattering, to obtain the *R*_*g*1_, *G*_1_, *D*_1_; (ii) The value of exponent *d*_1_ was fixed to 4.00 because it is independent of the applied pressures, as indicated by the plateaus in the *Iq*^4^-*q* plots of the USAXS–SAXS data (Figure 4); (iii) fix the parameters determined in step (i); and (iv) fit the high *q* region of 0.2 nm^−1^ to 7 nm^−1^ to extract the *R*_*g*2_, *G*_2_, *D*_2_, *d*_2_, and *B_g_*. The fitted parameters of *R*_*g*1_, *R*_*g*2_, and *d*_2_ are listed in Table 1 and Table 2. As the pressures increased from 1 kN to 30 kN, the *R*_*g*1_ decreased from 55.8 to 50.0 nm for NPD samples and from 38.2 to 30.2 nm for NND samples, while the *R*_*g*2_ (7.1 to 8.2 nm) kept nearly constant during compression. The exponents *d*_2_ were about 3 at low pressures (<5 kN) and gradually decreased to 2.8 for NND and 2.5 for NPD at 30 kN (Figure 5). Noted that all the experimental scattering curves exhibit an up-turn and deviate from the fitted curves at the smallest *q* region from 0.007 nm^−1^ to 0.02 nm^−1^ (Figure 2). This indicates the presence of even bigger voids with sizes larger than 300 nm (estimated by 2π over 0.02 nm^−1^), corresponding to the inter-granular space well visible in the SEM images (Figure 1). These structures were beyond the range accessible in the experiments, and were not included in the model analysis.

The Guinier–Porod model fitting over the *q* range 0.02 nm^−1^ to 7 nm^−1^ indicated the existence of two populations of voids in these samples. Interestingly, only the size of the big voids decreased with the applied pressures. This behavior suggests that the larger voids are located at the inter-granular space, which is more susceptible to deformation caused by the flow and fracture of the granules under die compaction. In theory, for a surface fractal, *d* varies between 3 to 4; for mass/volume fractal, 1 < *d* < 3; while for a sharp and smooth interface, *d* is 4 [35]. Based on the extracted values of surface fractal dimension, the large inter-granular voids in both TATB samples had smooth interfaces with the surrounding TATB granules, in good agreement with images obtained from electron microscopy (Figure 1). Compared with the NPD samples, the NND sample had smaller *R*_*g*1_, most likely as a result of smaller initial inter-granular spaces (Figure 1b). The *R*_*g*2_ of small voids was found to be smaller than that of the primary TATB. This result implies that the small voids correspond to intra-granular voids. The large inter-granular and small intra-granular voids in this work are in consistent with a reported SEM study [36]. At low pressures, the exponent *d* = 3 indicated a very rough interface between intra-granular voids and adjacent TATB crystals, and/or a polydispersity of the intra-granular voids with high connectivity, i.e., a ramified network structure [9,13,22,23]. At higher pressures, the exponent *d* became smaller than 3 (Figure 5), illustrating that the external pressure strongly affected the network structure of the intra-granular voids. That is, the volume-filling ratio of the network of voids became smaller at high pressures (>15 kN) than that at low pressures.

### 3.2. Porosity and Interfacial Area

The densities of the die pressed TATB discs at different pressures are listed in Table 3. As the pressures increased from 1 to 30 kN, the densities increased from 1.25 to 1.85 g/cm^3^ for NND samples and 1.07 to 1.83 g/cm^3^ for NPD samples. The total porosity *φ* of the die pressed nano-TATB discs was calculated by
(3)$$\varphi =1-\frac{d_{\mathrm{bulk}}}{d_{\mathrm{TATB}}}$$
where *d*_TATB_ is the density of TATB crystal (1.93 g·cm^−3^) [37]. The calculated porosities are listed in Table 3. The applied uniaxial die pressures led to a significant change in the porosity in the TATB discs, by an approximately ten-fold decrease of the calculated *φ* in both samples.

The porosity of the explosives visible by USAXS–SAXS in the size range accessible by the scattering experiment (*φ*_Q_) can be deduced by calculating the pseudo-invariant *Q*_pseudo_ [13,38], written as
(4)$$Q_{\mathrm{pseudo}}=\int_{0.007}^{7}q^{2}\frac{d\Sigma }{d\Omega }(q)dq=2\pi^{2}(\Delta \rho {)}^{2}\varphi_{Q}(1-\varphi_{Q})$$
where Δ*ρ* is the difference of scattering length density between void and TATB matrix. Due to the limited *q* range and the turn-up of intensities at lowest *q*, the total *φ* cannot be determined from the present scattering data. Therefore, the *φ*_Q_ with sizes roughly from 1 to 900 nm were calculated by limiting the integration range to the interval from 0.007 to 7 nm^−1^, following the approach used in studies on estimation of the porosities in RDX crystals and CaCO_3_ [13,39].

The porosities with sizes between 1 to 900 nm (*φ*_Q_) and larger than 900 nm (*φ* − *φ*_Q_) as a function of sample densities are displayed in Figure 6. The total volume of the voids decreased gradually with an increase in pressure, while volume of the voids with size below 900 nm decreased slowly at low pressures (<5 kN), and then rapidly at high pressures. Electron micrographs of the cross-sections of NPD and NND samples are shown in Figure 7. Both the samples presented plastic deformation behavior. With the increase of pressures, the TATB granules changed from ellipsoid to lamellar structure and the samples became more and more dense. The SEM results are in good agreement with the study of deuterated TATB using neutron diffraction [4]. They found that the nearly equiaxed TATB particles gradually fracture into plate-like particles, and the TATB (001) poles reorient preferentially in the pressing direction during consolidation.

The interfacial area is derived from Porod’s approximation [2]:(5)$$\;I\left(q\right)=2\pi \Delta \rho^{2}S_{\mathrm{Porod}}/q^{4}$$
where *S*_Porod_, proportional to the height of the plateaus in Figure 3, is the interfacial area between the inter-granular voids and the TATB matrix. Values of *S*_Porod_ and specific surface area (*S*_BET_), as measured by nitrogen adsorption are collected in Table 4. Both *S*_Porod_ and *S*_BET_ decreased with increasing pressures. *S*_BET_ was the total open surface area and *S*_Porod_ only reflected the interfacial area of the inter-granular voids with sizes of tens of nanometers, hence *S*_BET_ is larger than *S*_Porod_ at the corresponding pressures. Because of the network structure of NND samples, the initial *S*_BET_ of NND was larger than that of NPD. However, at high pressures, the *S*_BET_ of NND became smaller than that of NPD, indicating that the open voids of NPD collapsed due to the deformation of TATB crystals.

## 4. Discussion

In the present work, the combined use of SAXS and USAXS techniques was proven to be indispensable to probe the hierarchical structure of nanostructured TATB during compaction. The SAXS data provided the fractal network structure of the intra-granular voids, and the USAXS data provided size and interface information of the inter-granular voids. The scattering data, combined with SEM and nitrogen adsorption, provided an insight into the densification process of NPD and NND samples. At low pressures (<5 kN), the intra-granular voids with size larger 900 nm were sensitive to the pressures (Figure 6); while the *R*_*g*1_ remained constant (Table 1 and Table 2). These results suggest that the applied external forces primarily influence the inter-granular voids at a micron scale. The flow of the TATB granules under compression are the main consolidation process. At medium pressures (between 5 and 10 kN), the decrease of *R*_*g*1_ was accompanied by a decrease in void volume with size smaller than 900 nm. The *S*_Porod_ decreased from 2.1 to 1.3 m^2^/g for NND samples and from 2.5 to 1.3 m^2^/g for NPD samples as the pressures increased from 5 to 10 kN (Table 4). The loss of interfacial area, *R*_*g*1_, and porosity indicated that the flow, fracture, and plastic deformation of the TATB granules under compression were the main densification mechanisms at medium pressures. The collapse and filling of the inter-granular voids with size between tens of nanometer to submicron size governs the densification process. At high pressures (>15 kN), the exponent *d*_2_ decreased from 3 to 2.8 for NPD and 2.5 for NND at 30 kN. The severe plastic deformation of the layered TATB crystals at high pressures may have led to the collapse of the intra-granular voids, resulting in the decrease of *d*_2_ and a less volume-filling ratio of the ramified intra-granular voids.

In general, NPD and NND samples exhibited similar structural evolution upon uniaxial pressures. However, the pressure distribution was found to be more homogeneous in the NND samples. The *S*_Porod_ (4.1 m^2^/g) of NND was larger than that (3.0 m^2^/g) of NPD at 1 kN, and decreased to 0.6 m^2^/g for both the samples at 30 kN. The *S*_BET_ of NND and NPD was 15.8 and 9.3 m^2^/g at 1 kN, and decreased to 0.9 and 2.9 m^2^/g at 30 kN, respectively (Table 4). Compared to the NPD samples, the NND samples showed a characteristic transition from a porous structure to a compacted structure (Figure 7). The scattering and SEM data implied that the NND sample was more strongly influenced by the applied pressures than the NPD sample, due to its more uniform network structure.

In a series of investigations, Mang et al. [22,23] studied the microstructure of die pressed TATB powders in micron-scale by using SANS and USANS. Several notable differences between their studies and the present results are as follows: (1) the Guinier region (plateau) in *I-q* curves obtained from the nanostructured NND and NPD was identified at *q* > 0.01 nm^−1^, because of the smaller inter-granular voids (Figure 1 and Figure 5); (2) all voids in the consolidated TATB in the micron-scale were found to have rough fractal-like interfaces, while a smooth interface between inter-granular voids and TATB matrix was determined for NND and NPD samples; and (3) the intra-granular voids were not mentioned in their work, possibly because of the lower signal/noise ratio and lower sensitivity of the neutron experiments, while both the inter- and intra-granular voids could be well distinguished by USAXS–SAXS. However, both studies indicated that the response of the microstructure of die pressed TATB upon compaction is complex. The parameters obtained by using USAXS–SAXS can be used to relate the microstructure of consolidated TATB to its shock sensitivity and flame propagation, and are of interest from both safety and performance perspectives.

## 5. Conclusions

USAXS–SAXS techniques were used to characterize nanostructured TATB as a function of applied uniaxial pressure. The unified Guinier–Porod model analysis indicated the presence of inter- and intra-granular voids. As the pressures increased from 1 kN to 30 kN, the *R*_*g*1_ of the inter-granular voids decreased from 55.8 to 50.0 nm and from 38.2 to 30.2 nm for NPD and NND samples, respectively. The *R*_*g*2_ of the intra-granular voids was constant during compression. The fractal dimension *d*_2_ decreased from 3 at low pressures (<5 kN) to 2.8 (NPD) and 2.5 (NND) at 30 kN, indicating a less volume-filling ratio of the ramified intra-granular void. The Guinier–Porod model, pseudo-invariant, Porod’s law, densimetry, and SEM results showed that the flow of the TATB granules was the main consolidation process under low pressures (<5 kN); the flow, fracture, and plastic deformation of the TATB granules were the main densification mechanisms at medium pressures; and the collapse of the intra-granular void due to severe plastic deformation of the layered TATB crystals played a role at high pressures. Compared to the NPD samples, the NND samples showed a characteristic transition from a porous structure to a compacted structure, and the applied pressures influenced more strongly on NND samples due to their uniform network structure.

## Figures and Tables

**Figure 1 nanomaterials-13-00869-f001:**
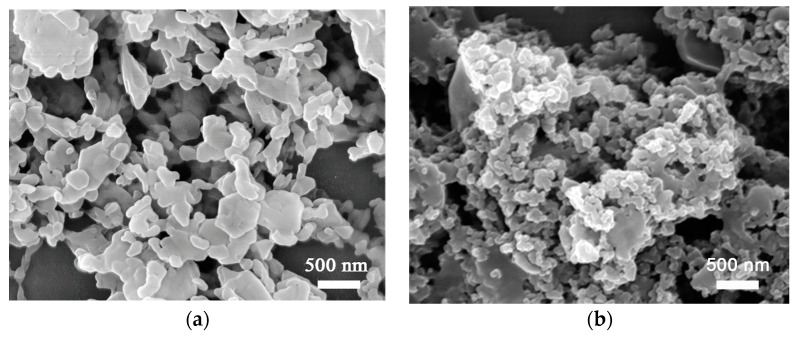
SEM micrographs of the initial TATB powders: (**a**) Nano-particle form; (**b**) Nano-networks form (**b**).

**Figure 2 nanomaterials-13-00869-f002:**
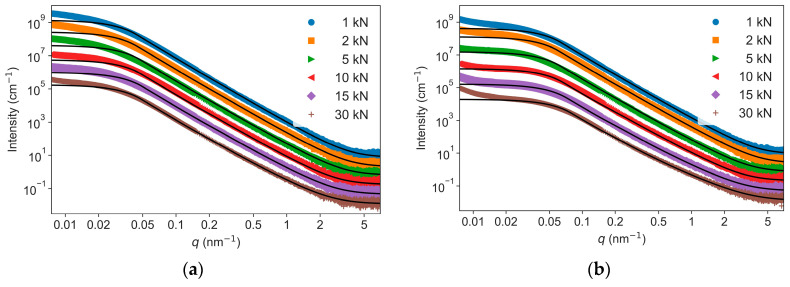
USAXS–SAXS data obtained from NPD (**a**) and NND (**b**). The curves are shifted vertically by factors of 4, 16, 64, 256, and 1024 for clarity (continuous black lines are the fits to a two-level Guinier–Porod model).

**Figure 3 nanomaterials-13-00869-f003:**
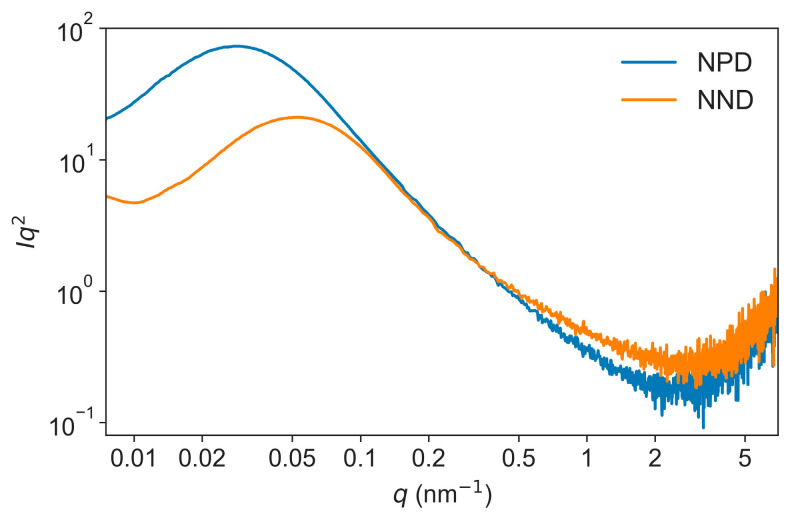
Kratky plot of the USAXS–SAXS data obtained from NPD and NND samples die pressed at 30 kN.

**Figure 4 nanomaterials-13-00869-f004:**
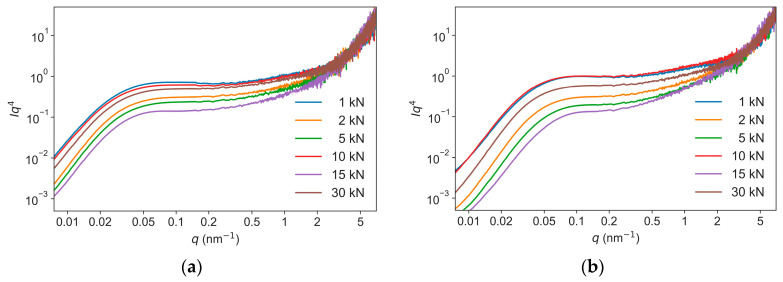
*Iq*^4^-*q* representations of the USAXS–SAXS data obtained from the NPD (**a**) and NND (**b**).

**Figure 5 nanomaterials-13-00869-f005:**
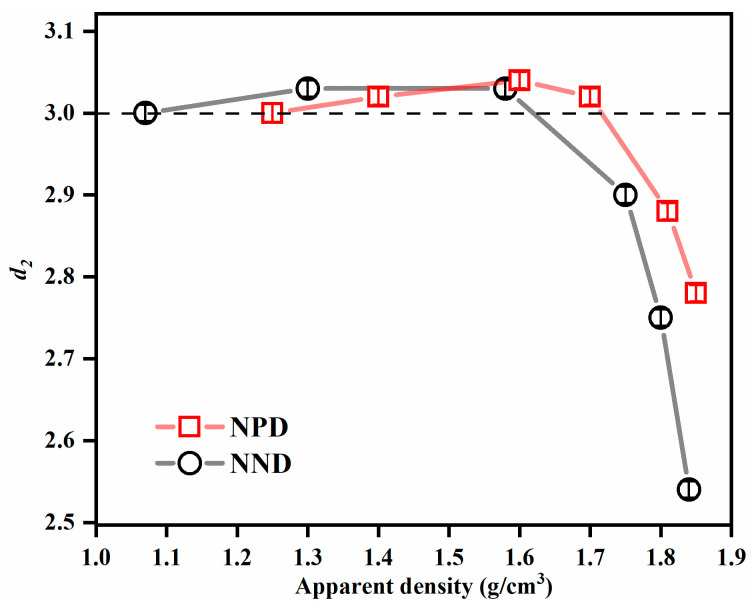
Fractal exponent derived from USAXS–SAXS data obtained from NPD and NND samples.

**Figure 6 nanomaterials-13-00869-f006:**
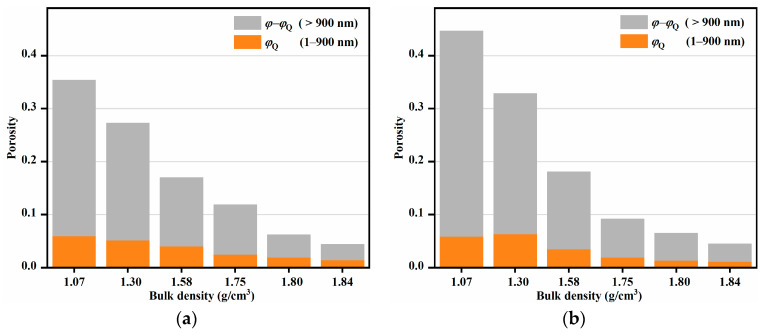
Porosities of NPD (**a**) and NND (**b**) as a function of density.

**Figure 7 nanomaterials-13-00869-f007:**
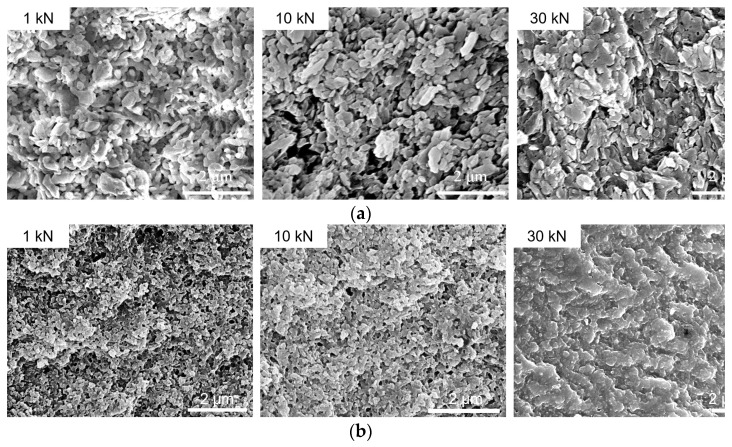
SEM images of the cross-section of NPD (**a**) and NND (**b**) die pressed at 1 kN, 10 kN, and 30 kN.

**Table 1 nanomaterials-13-00869-t001:** Structural parameters obtained from USAXS–SAXS data on NPD samples by least squares fitting of a two-level Guinier–Porod model.

Pressure (kN)	*R*_*g*1_ (nm)	*R*_*g*2_ (nm)	*d*_1_ (Fixed)	*d* _2_
1	55.8 ± 0.1	7.7 ± 0.1	4.00	3.00 ± 0.01
2	55.1 ± 0.1	7.7 ± 0.1	4.00	3.00 ± 0.01
5	52.7 ± 0.1	7.7 ± 0.1	4.00	3.00 ± 0.01
10	50.1 ± 0.1	8.2 ± 0.1	4.00	3.00 ± 0.01
15	49.8 ± 0.1	8.0 ± 0.1	4.00	2.90 ± 0.01
30	50.0 ± 0.1	7.8 ± 0.1	4.00	2.80 ± 0.01

**Table 2 nanomaterials-13-00869-t002:** Structural parameters obtained from USAXS–SAXS data on NND samples by least squares fitting of a two-level Guinier–Porod model.

Pressure (kN)	*R*_*g*1_ (nm)	*R*_*g*2_ (nm)	*d*_1_ (Fixed)	*d* _2_
1	38.7 ± 0.1	7.7 ± 0.1	4.00	3.00 ± 0.02
2	39.3 ± 0.1	7.1 ± 0.1	4.00	3.03 ± 0.01
5	38.9 ± 0.1	8.0 ± 0.1	4.00	3.03 ± 0.01
10	35.8 ± 0.1	8.1 ± 0.1	4.00	2.90 ± 0.01
15	33.1 ± 0.1	8.1 ± 0.1	4.00	2.75 ± 0.01
30	30.2 ± 0.1	8.0 ± 0.1	4.00	2.54 ± 0.01

**Table 3 nanomaterials-13-00869-t003:** Bulk densities (*d*_bulk_) and total porosities (*φ*) of the NPD and NND samples pressed at different pressures.

Pressure (kN)	*d*_bulk_ (g·cm^−3^)		*φ*	
	NPD	NND	NPD	NND
1	1.25	1.07	0.35	0.45
2	1.40	1.30	0.27	0.33
5	1.60	1.58	0.17	0.18
10	1.70	1.75	0.12	0.09
15	1.81	1.80	0.06	0.07
30	1.85	1.84	0.04	0.05

**Table 4 nanomaterials-13-00869-t004:** *S*_Porod_ derived from Porod’s approximation and *S*_BET_ measured by nitrogen adsorption isotherms of the NPD and NND samples treated at different pressures.

Pressure (kN)	*S*_Porod_ (cm^2^·g^−1^)		*S*_BET_ (cm^2^·g^−1^)	
	NPD	NND	NPD	NND
1	3.0	4.1	9.3	15.8
2	2.6	4.2	8.7	14.3
5	2.1	2.5	7.8	8.2
10	1.3	1.3	5.8	0.9
15	1.0	0.8	3.3	1.0
30	0.6	0.6	2.9	0.9

## Data Availability

Experimental data are available from the authors.

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
