# Peer review of "Densification of Two Forms of Nanostructured TATB under Uniaxial Die Pressures: A USAXS–SAXS Study"

_nanomaterials, 2023, doi:10.3390/nano13050869_

Round 1

Reviewer 1 Report

The authors should present their data and results in a more clear way.

- SAXS experimental data are reported in graphs without tics and without error bars;

-theoretical fitting curves of SAXS data are not reported;

-error bars should be explicitely written. Although the authors state that the error bar is less than the last digit, there is difference between 50.9 plus/minus 0.9 or 50.9 plus/minus 0.1.

The comparison between NPD and NND SAXS experimental curves should be reported, at least for a couple of investigated pressures. Why Kratky plots are not reported?

I think that the experimental data deserve an improved illustration, together with the punctual report of theoretical curves together with experimental points.

Reviewer 2 Report

see attachment

Round 2

Reviewer 1 Report

The authors modified data presentation according to my suggestion. I think that Kratky plots clearly evidence some differences in the samples which can be observed at first glance from not-SAXS experts. It improves the explanation of results.

Also error bars in the results have been added.

I am surprised that authors did not comment at all the bad fitting of USAXS data, at very low Q-values. I think it deserves at least a brief comment (in the previous version of the manuscript the theoretical fitting curves were not reported, and I could not appreciate this point).

I suggest to remove the word "obvious" from the manuscript, which is referred to the peak in the Kratky plots. Experimental results can be easily expected, but not obvious.

Author Response

We thank the Reviewer for the criticism and valuable suggestions. The word obvious has been avoided throughout the paper. The details of the larger structures, responsible for the upturn of the curves have been discussed in the revised version. The modifications are marked by red color.

Reviewer 2 Report

The paper is about densification of two forms of TATB under pressure and their study by USAXS-SAXS and is the revised version of the original one.

Now, I understand the novelty of the paper. The authors performed the requested corrections and I think that the paper was improved

So, the paper could be accepted for the publication in Nanomaterials

Author Response

We thank the Reviewer for the useful suggestions and for the appreciation of our results presented in the manuscript.